DUSP12 regulates the tumorigenesis and prognosis of hepatocellular carcinoma

Ju Gaoda 1
Zhou Tianhao 2
Zhang Rui 3
Pan Xiaozao 3
Xue Bing bingning0910@aliyun.com 3
Miao Sen miaosen128@163.com 3
1 Department of Medical Oncology, Beijing Cancer Hospital, Peking University , Beijing , China
2 Shanghai First People’s Hospital, Shanghai Jiao Tong University School of Medicine , Shanghai , China
3 Department of Pathology, Affiliated Hospital of Jining Medical University , Jining , China
Uversky Vladimir
Electronic publication date: 2021 Aug 3
Publication date: 2021
Volume: 9
Electronic Location ID: e11929
Received 2021 May 13; Accepted 2021 Jul 18
Copyright: ©2021 Ju et al.
Copyright year: 2021
Copyright holder: Ju et al.
License: This is an open access article distributed under the terms of the Creative Commons Attribution License, which permits unrestricted use, distribution, reproduction and adaptation in any medium and for any purpose provided that it is properly attributed. For attribution, the original author(s), title, publication source (PeerJ) and either DOI or URL of the article must be cited.
License URL: https://creativecommons.org/licenses/by/4.0/

Keywords: Hepatocellular carcinoma, DUSP12, Mutation, Tumorigenesis, Prognosis

Funding: The authors received no funding for this work.

==============================
Background

Dual specificity protein phosphatase (DUSP)12 is an atypical member of the protein tyrosine phosphatase family, which are overexpressed in multiple types of malignant tumors. This protein family protect cells from apoptosis and promotes the proliferation and motility of cells. However, the pathological role of DUSP12 in hepatocellular carcinoma (HCC) is incompletely understood.

Methods

We analyzed mRNA expression of DUSP12 between HCC and normal liver tissues using multiple online databases, and explored the status of DUSP12 mutants using the cBioPortal database. The correlation between DUSP12 expression and tumor-infiltrating immune cells was demonstrated using the Tumor Immune Estimation Resource database and the Tumor and Immune System Interaction Database. Loss of function assay was utilized to evaluate the role of DUSP12 in HCC progression.

Results

DUSP12 had higher expression along with mRNA amplification in HCC tissues compared with those in normal liver tissues, which suggested that higher DUSP12 expression predicted shorter overall survival. Analyses of functional enrichment of differentially expressed genes suggested that DUSP12 regulated HCC tumorigenesis, and that knockdown of DUSP12 expression by short hairpin (sh)RNA decreased the proliferation and migration of HCC cells. Besides, DUSP12 expression was positively associated with the infiltration of cluster of differentiation (CD)4+ T cells (especially CD4+ regulatory T cells), macrophages, neutrophils and dendritic cells. DUSP12 expression was positively associated with immune-checkpoint moieties, and was downregulated in a C3 immune-subgroup of HCC (which had the longest survival).

Conclusion

These data suggest that DUSP12 may have a critical role in the tumorigenesis, infiltration of immune cells, and prognosis of HCC.

Introduction

Hepatocellular carcinoma (HCC) is the most common malignant tumor of the liver (Ferlay et al., 2015). HCC is the third most prevalent cause of cancer-specific death worldwide (Bray et al., 2018). Due to rapid progression, HCC is usually discovered and diagnosed at an advanced stage, which leads to the loss of feasibility of treatments (Chen et al., 2016). Systemic chemotherapy for HCC is limited because HCC lacks sufficient targets for drugs, and HCC evolves resistance to classic anti-tumor agents (Fitzmorris et al., 2015; Lohitesh, Chowdhury & Mukherjee, 2018). The median duration of survival of patients with advanced HCC is ∼1 year (Llovet et al., 2018; Zhu et al., 2015).

Dual specificity protein phosphatase (DUSP)12 is an atypical member of the protein tyrosine phosphatase (PTP) family. DUSP12 regulates the proliferation, apoptosis, and migration of cells by dephosphorylating tyrosine and serine/threonine residues (Guan, Broyles & Dixon, 1991; Patterson et al., 2009). It has been reported that DUSP12 is overexpressed in intracranial ependymoma, retinoblastomas, and neuroblastomas (Gratias et al., 2005; Hirai et al., 1999; Mendrzyk et al., 2006). Some research teams have found that DUSP12 overexpression protects HeLa cells from apoptosis and promotes the proliferation and motility of HEK293 cells (Cain, Braun & Beeser, 2011; Sharda et al., 2009). Several studies have reported that DUSP12 overexpression in macrophages could reduce expression of proinflammatory cytokines such as tumor necrosis factor- α, interleukin (IL)-1 and IL-6, and increase IL-10 expression. DUSP12-expressed hepatocytes are less inflamed and cause less hepatic steatosis than DUSP12-deleted hepatocytes (Cho et al., 2017; Huang et al., 2019). However, the correlation between DUSP12 expression and HCC tumorigenesis and DUSP12 function in cells is not known.

We explored the expression, mutation, and pathological role of DUSP12 in HCC by integrated analyses of various data sources using online tools. The latter were applied to analyze the correlation of target genes with different cancer types, but especially HCC. In this way, we hoped to help researchers investigate the molecular targets of tumorigenesis.

Materials & Methods

Public databases

Ualcan

Ualcan (http://ualcan.path.uab.edu/) is an interactive Internet resource for analyzing cancer OMICS data. Ualcan was used to analyze transcription expression, prognosis, and the methylation of genes in The Cancer Genome Atlas (TCGA) datasets (Chandrashekar et al., 2017). The TCGA-Liver Hepatocellular Carcinoma (LIHC) dataset was employed in our research.

Gene expression profiling interactive analysis (GEPIA)

The GEPIA database (http://gepia.cancer-pku.cn/) was used to plot overall survival (OS) and disease-free survival (DFS) curves. Group cutoff was based on the median of gene expression and prognostic status of patients in the TCGA-LIHC dataset (Tang et al., 2017).

Kaplan–Meier Plotter (liver cancer)

Kaplan–Meier Plotter (http://kmplot.com/analysis/index.php/) was employed for OS (including 364 patients), DFS (including 316 patients), progression-free survival (PFS) (including 370 patients) and disease-specific survival (DSS) (including 262 patients) analysis using data from RNA-sequencing of a liver-cancer dataset (Menyhárt, Nagy & Gyorffy, 2018). We separated high and low expression based on the best cutoff expression value of DUSP12 that all possible cutoff values between lower and upper quartiles were computed, and the best performing threshold was used as a cutoff.

Human protein atlas (HPA)

The HPA database (http://www.proteinatlas.org/) was used to validate gene expression in liver-cancer tissues and normal liver tissues at the protein level (Uhlén et al., 2015; Uhlen et al., 2017).

Cancer cell line encyclopedia

The Cancer Cell Line Encyclopedia database (http://www.DepMap Broadinstitute.org/ccle/) was employed to analyze gene expression in HCC cells (Ghandi et al., 2019). RNA-expression data of liver-cancer cell lines were downloaded from this website for our research.

HCCDB

The Hepatocellular Cancer Database (HCCDB) (http://lifeome.net/database/hccdb/home.html) is an integrative molecular database of HCC with 15 datasets. Co-expressed genes were computed and displayed in HCC according the guidelines on the HCCDB website (Lian et al., 2018). The normalized expression data of ICGC-LIRI-JP and GSE14520 were downloaded from HCCDB.

cBioPortal

The cBioPortal database (http://cbioportal.org/) is an online tool for analyzing the mutation characteristics of genes in a Liver Hepatocellular Carcinoma (TCGA, Firehose Legacy) dataset (Cerami et al., 2012; Gao et al., 2013). A total of 392 differentially expressed genes (DEGs) were identified from a DUSP12-altered group and DUSP12-nonaltered group of patients.

Tumor immune estimation resource (TIMER)

The correlation of copy number variation (CNV) of genes with the abundance of tumor-infiltrating immune cells (TIICs) was displayed by an online tool in the TIMER database (Li et al., 2016; Li et al., 2017). In this way, we analyzed the correlation of gene expression with TICC abundance and expression of immune-checkpoint moieties in the TCGA-LIHC dataset (https://cistrome.shinyapps.io/timer/), the list of correlations was filtered for interactions with P <0.05 and correlation coefficient >0.2.

Tumor and immune system interaction database (TISIDB)

TISIDB (http://cis.hku.hk/TISIDB/) was employed to analyze gene expression in patients with different immune subtypes of HCC (Ru et al., 2019).

CIBERSORT

A total of 369 tumor samples extracted from the Genomic Data Commons (GDC)-TCGA-LIHC dataset were downloaded from UCSC.XENA (http://xena.ucsc.edu/). The CIBERSORT method was used within the R package (http://www.r-project.org/) (Goldman et al., 2020; Newman et al., 2019). After removing samples with P ≥ 0.05 in the result of CIBERSOT analysis, the remaining 261 samples were divided into high and low expression groups based on median expression of DUSP12. Bar graphs of the TIIC ratio between high and low expression groups were plotted with Prism 7 (GraphPad, San Diego, CA, USA).

Metascape

Metascape (http://metascape.org/) is a resource to aid the annotation and analyses of genes, which helps biologists make sense of one or multiple gene lists. Metascape was applied for analyses of protein–protein interaction (PPI) networks. Analyses of functional enrichment and enrichment of pathways were done using Gene Ontology (GO), Kyoto Encyclopedia of Genes and Genomes (KEGG) and DisGeNET databases (Zhou et al., 2019).

Depmap portal database

DepMap Portal (https://depmap.org/portal/) database is utilized to evaluate the probabilities of dependency of DUSP12 in HCC cell lines with CERES score based on data from CRISPR (DepMap 21Q2 Public + Score, CERES) cohort (DepMap Broad, 2021).

Cell lines and culture conditions

The human liver-cancer cell line Huh7 was acquired from American Type Culture Collection (Manassas, VA, USA) and cultivated in Dulbecco’s modified Eagle’s medium (DMEM; Gibco, Grand Island, NY, USA) with 10% fetal bovine serum (Gibco) and 1% penicillin–streptomycin (Gibco) at 37 °C in an atmosphere of 5% CO2.

Western blotting

Cell proteins were extracted by denaturing buffer and then quantified by a bicinchoninic acid protein assay (Thermo Scientific, Waltham, MA, USA). Protein lysates from the HCC cell line were separated by sodium dodecyl sulfate–polyacrylamide gel electrophoresis, transferred to nitrocellulose membranes (Millipore, Bedford, MA, USA), blocked, and then detected by primary antibody DUSP12 (1:2000 dilution; catalog number: ab237008; Abcam, Cambridge, UK) and horseradish peroxidase-conjugated secondary antibody (Sigma–Aldrich, Saint Louis, MO, USA). These actions were followed by exposure to enhanced chemiluminescence. The housekeep gene β-tubulin (1:500; ab6046; Abcam) was used as a loading control.

Plasmids and lentivirus production

Annealing and connection of short hairpin (sh)RNA were undertaken followed by construction into the modified plasmid pLKO.1. Well-constructed vectors were transinfected into HEK293T cells by lentivirus packaging plasmids psPAX and pMD2.0G. The shRNA sequences and DUSP12 primer sequence (forward and reverse, respectively) were: CCGGGTTGAGTGGCAACTGAAATTATCTCGAGATAATTTCAGTTGCCACTCAAGTTTTTG, and AATTCAAAAAGTTGAGTGGCAACTGAAATTATCTCGAGATAATTTCAGTTGCCACTCAAG for shDUSP12-1; CCGGGTGGATACCTCTAGTGCAATTCTCGAGAATTGCACTAGAGGTATCCACTTTTTG, and AATTCAAAAAGTGGATACCTCTAGTGCAATTCTCGAGAATTGCACTAGAGGTATCCAC for shDUSP12-2.

Cell-growth assay

Lentivirus-infected stable cells were seeded into 96-well plates and cultured in DMEM containing 10% fetal bovine serum (2000 cells per well, five parallel wells). Then, cells were collected at different timepoints. The cell number in each well was counted by Cell Counting Kit 8 (CCK8). Absorbance at 450 nm was measured to determine of the number of viable cells.

Transwell™ assay

Lentivirus-infected stable cells were seeded in the upper chamber of a Transwell chamber (24-well (8-µm pore; Corning, Corning, NY, USA) in 200 µL of serum-free DMEM (1 ×105 cells per well, five parallel wells). Then, 800 µL of DMEM containing 10% fetal bovine serum was added to the lower chamber and incubation allowed for 36 h at 37 °C. After removing the cells at the upper surface of the membrane, cells were passed through a filter and fixed with 4% paraformaldehyde, stained with 0.1% Crystal Violet solution and photographed using an inverted fluorescence microscope.

Results

Pattern of DUSP12 transcriptional expression using Ualcan and HCCDB databases

Analyses of the TCGA-LIHC dataset in the Ualcan database and analyses of ICGC-LIRI-JP and GSE14520 cohorts revealed that transcriptional expression of DUSP12 was higher in LIHC tissues compared with that in normal liver tissues (Fig. 1A, Fig. S1). Analyses of clinical subgroups demonstrated that DUSP12 expression was higher in an Asian, tumor–node–metastasis (TNM) stage-III, grade-3, P53-mutant group than that in a Caucasian, TNM stage-I/II, grade-1/2, P53-nonmutant group. The non-significant change in DUSP12 expression between the stage-IV/grade-4/N1 group compared with that in other groups may have been caused by the number of samples in the stage-IV/grade-4/N1 group being significantly less compared with that in other groups. The top positively and negatively correlated genes in TCGA dataset for DUSP12 were also downloaded from UALCAN database (Figs. S1 and S2). We suspected that DUSP12 could be a pathological and prognostic marker of LIHC (Figs. 1B–1F). Integrative analyses of HCCDB revealed that mRNA expression of DUSP12 was higher in liver-cancer tissue compared with that in normal liver tissue in 11 cohorts (Fig. 2A). Analyses of functional enrichment of genes co-expressed with DUSP12 in liver-cancer tissues using Metascape showed that these genes were engaged mainly in “histone methylation”, “cullin RING ubiquitin ligase complexes”, “nuclear specks” and “ubiquitin-like protein transferase activity” (Figs. 2B–2E).

Figure 1 DUSP12 expression in the Ualcan database

(A) Normal vs. primary tumor. (B) Ethnicity. (C) Stage. (D) Grade. (E) Nodal metastasis. (F) TP53 mutation. ∗P < 0.05, ∗∗P < 0.01, ∗∗∗P < 0.001, ∗∗∗∗P < 0.0001.

Figure 2 Analysis of gene expression in the HCCDB

(A) DUSP12 expression in 11 cohorts. (B) PPI network containing DUSP12. (C) Heatmap for selected genes in the GO database (P < 0.05). (D) Network colored by cluster. (E) Network colored by P-value. GO, gene ontology; PPI, protein-protein network.

Validation of DUSP12 expression in tissues at the protein level and liver-cancer lines

DUSP12 expression was higher in liver-cancer tissues than that in normal liver tissues at the protein level, and this observation was validated by results from the HPA database (Antibody: HPA008840; cancer-patient ID: 2399; normal-patient ID: 3222) (Figs. 3A, 3B) and HUH1 had the highest transcriptional expression of DUSP12 in liver-cancer cell lines (Fig. 3C).

Figure 3 DUSP12 expression in liver-cancer tissues, normal liver tissues, and liver-cancer cell lines.

(A) Normal liver tissues. (B) Liver-cancer tissues. (C) Liver-cancer cell lines.

Survival analyses of liver-cancer patients with different DUSP12 expression

LIHC patients with higher expression of DUSP12 had shorter OS and DFS in the TCGA-LIHC dataset (Figs. 4A, 4B). Next, we validated the result by survival analyses of liver-cancer patients in the Kaplan–Meier Plotter database. We showed that patients with higher expression of DUSP12 had shorter OS, DFS, PFS and DSS than that of patients with lower expression of DUSP12 (Figs. 4C–4F).

Figure 4 Survival analyses of HCC patients with high expression of DUSP12 and low expression of DUSP12.

(A) OS in GEPIA. (B) DFS in GEPIA. (C) OS in Kaplan–Meier Plotter. (D) DFS in Kaplan–Meier Plotter. (E) PFS in Kaplan–Meier Plotter. (F) DSS in Kaplan–Meier Plotter. OS, overall survival; DFS, disease free survival; PFS, progression free survival; DSS, disease specific survival; GEPIA, Gene Expression Profiling Interactive Analysis.

Analyses of DUSP12 expression using the cBioPortal database

An online tool in the cBioPortal database was utilized to analyze the mutant status of DUSP12 in the Liver Hepatocellular Carcinoma (TCGA, Firehose Legacy) dataset. The mutant frequency of DUSP12 in HCC was 33.0% (Fig. 5A), which was composed mainly of amplification and high expression of mRNA (Fig. 5B). mRNA expression of DUSP12 in HCC with amplification was higher than that in those without alteration (Fig. 5C). HCC patients with altered DUSP12 had shorter OS than HCC patients with nonaltered DUSP12 (Fig. 5D). CNV analyses revealed that mRNA expression of DUSP12 was higher in HCC patients with DUSP12-amplification and DUSP12-gain patients than in those with DUSP12-shallow deletion and DUSP12-diploid (Fig. 6A). In general, mRNA expression of DUSP12 was correlated negatively with methylation of the promoter region of DUSP12 in 359 HCC samples from the TCGA-LIHC (Firehose Legacy) dataset (Fig. 6B). DUSP12-altered patients had a higher serum level of alpha fetoprotein (AFP) at procurement (Fig. 6C), fraction of genome altered (Fig. 6D) and worse histology grade in neoplasms (Fig. 6E). In addition, we screened 392 DEGs between DUSP12-altered patients and DUSP12-nonaltered patients with false discovery rate <0.05 and —log(ratio)—>1 (Fig. 7A). Analyses of functional enrichment and disease-related enrichment of these 392 DEGs revealed that these genes mainly took part in: “M61392: CHIANG LIVER CANCER SUBCLASS PROLIFERATION DN”; “M16496: CHIANG LIVER CANCER SUBCLASS CTNNB1 UP”; “M3268: CHIANG LIVER CANCER SUBCLASS PROLIFERATION UP” (Figs. 7B, 7C). Minimal Common Oncology Data Elements (MCODE) analyses revealed clustering of seven MCODEs (Fig. 7D). MCODE 1 mainly included proteins that took part in the cell cycle (e.g., cluster of differentiation (CD)K1, CDC20, PLK1), cell division or mitosis (e.g., AURKB, BIRC5, KIF2C, CENPA, CENPF). MCODE 2 mainly included cytochrome P450 (CYP) monooxygenase proteins that took part in various types of metabolism. MCODE 3 mainly included UDP-glucuronosyltransferase (UDPGT) proteins. MCODE 4 included GCK, OTC, SDS, POLE2 and ATP6V1B1. MCODE 5 included SULT4A1, SULT1B1, HS6ST2, and GAL3ST1. MCODE 6 mainly included C-C motif chemokine proteins. MCODE 7 included FMO3, ACSL6 and HSD11B1. MCODE 1 and MCODE 2 may play a critical part in the proliferation and biological activity of HCC cells. MCODE 6 may have a correlation with the infiltrations of immune cells in HCC samples.

Figure 5 Analyses of mutant status of DUSP12 in HCC.

(A) Frequency of the DUSP12 mutation. (B) Frequency of mutant types. (C) mRNA expression of DUSP12 in HCC cases with various types of mutant status. (D) OS of DUSP12-altered and DUSP12-nonaltered groups. OS, overall survival.

Figure 6 Clinical features of HCC patients with or without alteration of DUSP12 expression.

(A) mRNA expression of DUSP12 in HCC patients with different copy numbers. (B) Relationship between mRNA expression of DUSP12 and DUSP12 methylation. (C) AFP level at procurement in DUSP12-altered and DUSP12-nonaltered groups. (D) Fraction of genome altered in DUSP12-altered and DUSP12-nonaltered groups. (E) Histology grade in DUSP12-altered and DUSP12-nonaltered groups. AFP, alpha-fetoprotein; ∗P < 0.05, ∗∗∗P < 0.001.

Figure 7 DEGs between HCC cases with altered and nonaltered DUSP12.

(A) Volcano plot of DEGs between HCC cases with altered and nonaltered DUSP12. (B) Heatmap for selected terms (P < 0.05). (C) Network colored by cluster. (D) MCODEs of the PPI network. DEGs, differentially expressed genes; MCODEs, Minimal Common Oncology Data Elements; PPI, protein-protein network.

Analyses of DUSP12 expression using the TIMER database

We explored the correlation of DUSP12 expression with TIICs by TIMER database. DUSP12 expression was moderately (partial correlation >0.2) positively correlated with the abundance of infiltrating B cells, CD4+ T cells, macrophages, neutrophils, and dendritic cells (Fig. 8A). Then, we explored the correlation between DUSP12 and immune cells with other immune-infiltration analyzing methods (xCELL, EPIC, CIBERSORT). The result suggested that DUSP12 expression was moderately (partial correlation >0.2) positively correlated with the abundance of infiltrating B cells and CD4+ T cells regardless of the method employed (Table 1). DUSP12 expression had a moderately positive correlation with the immune-checkpoint moieties HAVCR2, TIGIT, CTLA4 and PD-1 (Fig. 8B). The infiltration level of CD8+ T cells, macrophages, neutrophils, and dendritic cells was significantly different in DUSP12 with different CNV (Fig. 8C). Furthermore, we explored mRNA expression of DUSP12 in patients with different immune-subgroup liver cancer using the TISIDB. We revealed that the C1 (wound healing) subgroup had the highest DUSP12 expression, whereas the C3 (inflammatory) and C6 (TGF-β dominant) subgroups had lower expression of DUSP12 (Fig. 8D). Furthermore, groups with high expression of DUSP12 had a higher ratio of T-regulatory cells (Tregs) and activated natural-killer cells compared with those with low expression of DUSP12 (Fig. 9).

Figure 8 Correlation between DUSP12 expression and number of tumor-infiltrating immune cells.

(A) Pearson correlation between DUSP12 expression with TIICs abundance. (B) Pearson correlation between DUSP12 expression with immune-checkpoint moieties. (C) DUSP12 expression in HCC cases with different copy numbers. (D) DUSP12 expression in HCC cases with different immune subtypes. TIIC, tumor infiltrating immune cells. ∗P < 0.05, ∗∗P < 0.01.

Table 1 The correaltion of DUSP12 and immune cell with four differernt immune-infiltration analyzing methods (adj. p < 0.05).

Tools	infiltrates	rho	p	adj.p	
xCELL	B cell memory_XCELL	0.144057569	0.0073615	0.043337621	
B cell_XCELL	0.199439884	0.000192613	0.002495049	
Class-switched memory B cell_XCELL	0.184646391	0.000567075	0.005704576	
Macrophage M2_XCELL	−0.514075954	1.15E−24	3.30E−21	
Macrophage_XCELL	−0.242486638	5.22E−06	0.000115083	
Myeloid dendritic cell activated_XCELL	0.197596364	0.00022132	0.002791875	
Plasmacytoid dendritic cell_XCELL	−0.148823775	0.005611022	0.036069056	
T cell CD4+ memory_XCELL	0.196441351	0.000241293	0.002953574	
T cell CD4+ Th2_XCELL	0.360960796	4.69E−12	5.17E−10	
TIMER	B cell_TIMER	0.411173294	1.66E−15	3.66E−13	
Macrophage_TIMER	0.337986628	1.15E−10	8.78E−09	
Myeloid dendritic cell_TIMER	0.476711143	5.62E−21	4.03E−18	
Neutrophil_TIMER	0.246221689	3.69E−06	8.33E−05	
T cell CD4+_TIMER	0.298486339	1.57E−08	6.73E−07	
EPIC	B cell_EPIC	0.162734334	0.002430024	0.019352438	
Macrophage_EPIC	−0.480988446	2.24E−21	2.14E−18	
T cell CD4+_EPIC	0.151948975	0.004676291	0.031694861	
CIBERSORT	B cell memory_CIBERSORT-ABS	0.143292145	0.007684113	0.044686314	
B cell plasma_CIBERSORT-ABS	0.154562661	0.004005	0.027869748	
Macrophage M0_CIBERSORT	0.180630369	0.000749909	0.007003218	
Macrophage M0_CIBERSORT-ABS	0.305859665	6.63E−09	3.22E−07	
Macrophage M1_CIBERSORT-ABS	0.314940048	2.21E−09	1.19E−07	
Macrophage M2_CIBERSORT-ABS	0.349115494	2.52E−11	2.26E−09	
Myeloid dendritic cell resting_CIBERSORT	0.178434015	0.000871612	0.008087095	
Myeloid dendritic cell resting_CIBERSORT-ABS	0.21779062	4.51E−05	0.00073801	
T cell CD4+ memory resting_CIBERSORT-ABS	0.269145392	3.88E−07	1.10E−05	
T cell CD8+_CIBERSORT-ABS	0.186805703	0.000486791	0.005056632	

Figure 9 Ratio of various TIICs in HCC.

∗P < 0.05. TIICs, tumor infiltrating immune cells.

Knockdown of DUSP12 expression decreases the proliferation and migration of Huh-7 cells

We evaluated the probabilities of dependency of DUSP12 in 22 HCC cell line types with data from CRISPR (DepMap 21Q2 Public + Score, CERES) cohort. The CERES scores of cell lines ranged from −0.04 to −0.47 while mean value was equal to −0.21 (Fig. S2). In general, a lower score meant that a gene is more likely to be essential in a given cell line and a score <0 meant that down-regulation of a gene may inhibit the proliferation of a given cell line. Human liver-cancer cells (Huh-7) were transfected with a specific shRNA for DUSP12 (shDUSP12) and a nonspecific shRNA (NC) (Fig. 10A). CCK8 and Transwell assays were utilized to evaluate the ability of cells to proliferate and migrate. Knockdown of DUSP12 expression led to the reduced proliferation and motility of Huh7 cells (Figs. 10B, 10C).

Figure 10 Knockdown of DUSP12 expression reduces the proliferation and migration of Huh7 cells.

(A) Knockdown of DUSP12 expression in Huh7 cells. (B) Proliferation of cells according to the CCK8 assay. (C) Migration of cells according to the Transwell™ assay. ∗∗∗∗P < 0.0001.

Discussion

We propose that expression of DUSP12, a member of the PTP family, was different in HCC tissues and normal liver tissues in multiple datasets. In addition, the clinical features of HCC patients had a strong relationship with DUSP12 expression, including ethnicity, TNM stage, histology grade, and P53-mutant status. Survival analyses using the Kaplan–Meier method demonstrated that HCC patients with higher expression of DUSP12 had shorter survival than those with lower expression of DUSP12.

A PPI network containing DUSP12 and 20 genes in liver-cancer samples in HCCDB was constructed. These genes, including ubiquitin-conjugating enzyme E2T (UBE2T), ILF2, SETDB1, CCT3, and UFC1, were engaged mainly in histone methylation, cullin RING ubiquitin ligase complexes, nuclear specks, and ubiquitin-like protein transferase activity. UBE2T has been demonstrated to promote the growth of HCC cells by regulating ubiquitination of P53, transition of the G2/M phase of the cell cycle, and the protein kinase B signaling pathway (Liu et al., 2019a; Liu et al., 2017; Wei et al., 2019). IL-F2 is a transcription factor which regulates the growth of HCC cells by controlling mRNA expression of apoptosis-related proteins (Cheng et al., 2016). Setdb1 is a histone methyltransferase that also regulates the growth of HCC cells by P53 methylation (Fei et al., 2015). CCT3 shows high expression in liver cancer, and leads to short survival (Liu et al., 2019b). CCT3 triggers expression of YAP and TFCP2 to regulate HCC tumorigenesis (Liu et al., 2019b). Hence, DUSP12 may have a critical role in the processes mentioned above by interacting with proteins in this network.

The mutant status of DUSP12 in HCC patients was determined by utilizing an online tool in the cBioPortal database. We discovered that nearly one-third of HCC patients suffered a DUSP12 mutation. Most of them experienced amplification and higher mRNA expression of DUSP12. HCC patients with a DUSP12 mutation had shorter survival, higher serum level of AFP, and worse histology grade than those of patients with wild-type DUSP12. Taken together, these findings suggest the probability of an intimate correlation between DUSP12 mutation and the pathology and prognosis of DUSP12 in HCC.

We also identified 392 DEGs between HCC patients with altered DUSP12 and HCC patients with nonaltered DUSP12. These DEGs mainly regulated the tumorigenesis and proliferation of HCC cells. MCODE analyses revealed clustering of seven MCODEs. Among them, MCODE 1 mainly comprised the mutations of CDK1, KIF2C, KIF18A, CENPA, and PLK1, which take part in the tumorigenesis and progression of liver cancer (Jung et al., 2019; Komatsu et al., 2009; Li et al., 2020; Li et al., 2011; Long et al., 2018; Luo et al., 2018; Wu et al., 2018; Zhang et al., 2015). MCODE 2 was composed mainly of members belonging to the CYP enzyme superfamily, which has a critical role in drug metabolism or chemical metabolism in the liver (Agundez, 2004). Agundez and colleagues found that the activity of CYP enzymes was closely associated with the risk of liver cancer (Agundez, 2004). MCODE 3 mainly included UDPGT proteins that take part in conjugation and subsequent elimination of potentially toxic xenobiotics and endogenous compounds. With regard to the components of MCODE 4, GCK helps to facilitate the uptake and conversion of glucose by acting as an insulin-sensitive determinant of hepatic-glucose usage (Velho et al., 1996) and OTC catalyzes the second step of the urea cycle (Horwich et al., 1984). With respect to the components of MCODE 6, it has been reported that CCL20 facilitates Treg activity in advanced HCC (Li & Liu, 2016).

We explored the association between TICCs and DUSP12 expression. DUSP12 expression was positively correlated with the abundance of tumor-infiltrating CD4+ T cells, macrophages, neutrophils, dendritic cells, and expression of the immune-checkpoint moieties HARVC2, TIGIT, CTLA4 and PDCD1. Increased expression of these immune-checkpoint moieties denoted a phenotype of liver cancer associated with a poor outcome. Use of TISIDB revealed that patients in the C3 immune-subgroup had the longest survival (Thorsson et al., 2018) and had the lowest expression of DUSP12 (except the C6 group, which contained only one patient). Furthermore, we investigated the ratio of various types of immune cells in total TIICs between a DUSP12-high-expression group and DUSP12-low-expression group. We showed that the infiltrating abundance of Tregs was higher in DUSP12-high-expression HCC samples compared with that in DUSP12-low-expression HCC samples. Tregs can inhibit the anti-tumor effects of immune cells and facilitate immune evasion by liver-cancer cells (Jiang et al., 2017; Langhans et al., 2019). This phenomenon may be one of the reasons why patients with high expression of DUSP12 experience rapid progression of disease and have a shorter survival time.

Conclusions

We propose that DUSP12 has a critical role in the tumorigenesis and progression of HCC. DUSP12 could be a potential target for curing liver cancer.

Supplemental Information

Supplemental Information 1 DUSP12 expression in HCC and normal liver tissues in ICGC-LIRI-JP and GSE14520 cohorts

DUSP12 expression in HCC and normal liver tissues in ICGC-LIRI-JP and GSE14520 cohorts. (A) ICGC-LIRI-JP. (B) GSE14520. ****P < 0.0001.

Click here for additional data file.

Supplemental Information 2 The top positively correlated genes in TCGA dataset for DUSP12 (Pearson correlation coefficient > = 0.3)

The top positively correlated genes in TCGA dataset for DUSP12 (Pearson correlation coefficient > = 0.3).

Click here for additional data file.

Supplemental Information 3 Dependency of DUSP12 in HCC cell lines

Dependency of DUSP12 in HCC cell lines

Click here for additional data file.

Supplemental Information 4 The top negatively correlated genes in TCGA dataset for DUSP12 (Pearson correlation coefficient < = −0.3)

The top negatively correlated genes in TCGA dataset for DUSP12 (Pearson correlation coefficient < = −0.3).

Click here for additional data file.

Supplemental Information 5 Raw data of CCK8 assay

Click here for additional data file.

Supplemental Information 6 Raw data (WB-internal control)

Click here for additional data file.

Supplemental Information 7 Raw data (WB-DUSP12)

Click here for additional data file.

Supplemental Information 8 Cell couts of transwell assay (imageJ)

Click here for additional data file.

The results of this study are based on the online databases Ualcan, GEPIA, HCCDB, Kaplan–Meier Plotter, TIMER, HPA, CCLE, TISIDB, CIBERSORT, cBioPortal, Metascape as well as DepMap. We thank the contributors who provided these databases/resources.

Additional Information and Declarations

Competing Interests

Author Contributions

Data Availability

The authors declare there are no competing interests.

Gaoda Ju conceived and designed the experiments, analyzed the data, prepared figures and/or tables, authored or reviewed drafts of the paper, and approved the final draft.

Tianhao Zhou conceived and designed the experiments, performed the experiments, prepared figures and/or tables, authored or reviewed drafts of the paper, and approved the final draft.

Rui Zhang performed the experiments, prepared figures and/or tables, and approved the final draft.

Xiaozao Pan analyzed the data, prepared figures and/or tables, and approved the final draft.

Bing Xue analyzed the data, authored or reviewed drafts of the paper, and approved the final draft.

Sen Miao conceived and designed the experiments, analyzed the data, authored or reviewed drafts of the paper, and approved the final draft.

The following information was supplied regarding data availability:

The CCK8 data, Transwell data, and western blots are available in the Supplemental Files.

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
