# Peer review of "DUSP12 regulates the tumorigenesis and prognosis of hepatocellular carcinoma"

_PeerJ, doi:10.7717/peerj.11929_

## Round 0.1 · original submission · Major Revisions

Please address the critiques of all reviewers and amend your manuscript accordingly.

Reviewer 1 ·

Basic reporting

This article is trying to elucidate the role of DUSP12 gene and hepatocellular carcinoma.

Experimental design

The overall design of this study is comprehensive. The report of the methods is robust.

Validity of the findings

The validity of the results is reliable. The findings and conclusions were well supported.

Additional comments

This article is trying to elucidate the role of DUSP12 gene and hepatocellular carcinoma. The overall design of this study is comprehensive. The report of the methods is robust. The validity of the results is reliable. The findings and conclusions were well supported. Thus, I suggest accepting this paper.
Figure 4, for survival analysis, please specify how do you separate high and low expression? Please also add a sample number for each group in figure 4C-F.
1. Please increase the font size in figure 1, 5, 6, and 8.
2. Figure 8a and 8b, please set a threshold for correlation analysis, combine correlation coefficient and P values.
3. Please transfer figure S3A, S3B, and S3c to figure 9 or 10.

·

Basic reporting

Ju et al, propose the role of DUSP12 in the pathogenesis of HCC. The authors use an extensive and data-rich approach to support their findings. I find the data and the model of high clarity. I would support the publication of this interesting concept if the authors could address some important comments.

First, the language used in clear, a bit harsh in some sections. The authors discuss the existing literature satisfactory.

Experimental design

Although the link between DUSP12 and HCC is biased (not supported by an unbiased retrospective approach), analyses used are well controlled, explained and extensively described. I would like the authors to clarify if the p values in figure 8 are adjusted or not.

I have a couple of additional suggestions experimental-wise that will be explained in the following section.

Validity of the findings

Although the model proposed from the authors is solid and thorough, I would like to propose some additional analyses.

1. The authors could analyze the cell line dependency to DUSP12, using the Achilles dependency project of the Broad institute https://depmap.org/portal/
if this model shows dependency of HCC cell lines to DUSP12 in an unbiased way, it will go a long way on supporting the proposed model

2. The authors could also use SNE plots to show the cluster that express DUSP12 in their scRNA seq data.

3. It would be extremely interesting to include a table with the top positively and negatively correlated genes in TCGA dataset for DUSP12.

Reviewer 3 ·

Basic reporting

Reporting looks to be in good form.

Experimental design

The connection between DUSP12 and HCC is carefully explored in this paper and represents original primary research suitable for this journal.

Validity of the findings

Analyses were conducted with multiple modalities -- multiple online databases and loss of function assays.

Conclusions are well stated -- not overly stated -- and limited to supporting results.

Additional comments

A comprehensive analysis was conducted investigating the basic science and translational properties of DUSP12 as it relates to HCC. This seems to be a meticulously prepared research report and I do not have any criticisms.

·

Basic reporting

In the manuscript (#61090) entitled, “DUSP12 regulates the tumorigenesis and prognosis of hepatocellular carcinoma”, the authors have have aimed to evaluate the role of DUSP12 in tumorigenesis, infiltration of immune cells, and prognosis of hepatocellular carcinoma (HCC). Ju et al have utilized a variety of online tools, multiple online databases, and loss of function assay to evaluate the role of DUSP12 in HCC. The authors however have failed to use a well-planned and comprehensive approach to come up with a logical conclusion and the manuscript is not well written.
The data presented/experiments performed are not rigorous enough and does not justify the conclusions.
The major drawback of the manuscript is that it is written very vaguely, making it difficult to comprehend. Authors in many places through the manuscript have just written down speculations without any concrete evidence and have failed to comprehensively explain their own findings.
Figure legends merely indicate the technique/graph and are not informative or comprehensive at all.
A very important issue about this manuscript is the language. There are numerous grammatical errors, typographical errors, punctuation mistakes, confusing sentences and a peculiar choice of words in many instances making the comprehension of data difficult. Hence the authors should revisit the manuscript and edit it thoroughly preferably with the help of someone with command in English language.

Experimental design

Results are just an extension of Materials and Methods and do not explain the findings.
Discussion is merely a summarization of results and hardly manages to explain the findings making it extremely difficult to the reader to comprehend the relevance/rationale of the findings and techniques used.
The name of the cell line Huh7 is represented in the manuscript differently in different places (Huh-7), needs homogenization. Also, can the authors comment on using Huh7 cell line for their experiments, although as per their data (Figure 3C) it is on the lower end of spectrum for DUSP12 expression.
There are a lot of question marks regarding the data throughout the manuscript which I have tried to detail, hence, making it impossible to the reviewer to be confident about the findings put forward by the authors.

Validity of the findings

Figure 1C,D,E: The number of samples in stage 4/grade 4/N1 are significantly less as compared to others and hence there is no significant change in those, authors fail to acknowledge/explain this.
Figure 3A,B: No mention of the technique used to get these images/results anywhere in the manuscript.
Figure 6B: No mention of number of samples/replicates.
Figure 7D: MCODE analyses, the reviewer is unclear about the relevance and significance of this and the authors have made no attempt in the manuscript to comprehensively explain/justify it.
The authors need to revisit their data and try to make logical conclusions in a more comprehensive manner based on their original findings rather than just putting down speculations.

Additional comments

1. Formatting errors throughout the manuscript (even in sub-headings)- Line 48 (function not functional), Line 100 (TNF-alpha?), Line 223 (protein), Line 264 (TGF-beta?), Line 268 (different), Line 277 (motility).
2. Spacing errors Line 86, 163, 239.
3. Grammatical errors Line 89 (and is?), Line 103-104, Line 132 (was downloaded), Line 161 (provides), Line 164 (were utilized), Line 173 (Huh7 was), Line 205 (After removed the cells?).
4. Line 90-91: “The median duration of survival of patients with advanced HCC is 1 year, do the authors mean Chinese population? as the reference used is based on data from a Chinese medical center (Zhu et al 2015).
Line 95-96: Order in which the references are cited is incorrect.
Line 153: A total of 369 tumor samples extracted from the GDC, Line 155 A total of 130 of 261 tumor samples, unclear?
Line 247: 392 DEGs mentioned, where as Line 141 mentions only 369.
Line 269-272: The sentence is unclear and does not make sense in its current form, needs formatting.
Line 282: The authors propose this not infer as the manuscript does not have confirming data, also the role of DUSP12 has been previously mentioned in the manuscript and hence redundant here.
Line 298: Again the statement is more of a speculation than conclusion of results.
Line 313: MCODE analyses, authors justb explain 1 and 2, with no justification/mention of others.
Line 325: We showed that Uusp12 could recruit Tregs to infiltrate liver-cancer tissues, how, where is the data/explanation to this?

---

## Round 0.2 · accepted · Accept

All critiques were addressed and the manuscript was amended accordingly. Therefore revised version is acceptable now.

Reviewer 1 ·

Basic reporting

No new comments.

Experimental design

No new comments.

Validity of the findings

No new comments.

Additional comments

No new comments.

·

Basic reporting

The changes made by the authors in regard to the comments by reviewers has made the manuscript more comprehensible than before and has improved the general writing and language. Results are clearer now.

Experimental design

New changes to the methods/results section makes the experimental approach clearer and understandable to the reader.

Validity of the findings

The results of the study are beneficial for the field and concluded correctly.

Additional comments

The editing has made the manuscript much better and more comprehensible. I am happy with the revision.